# Antihypertensive drug use and prostate cancer-specific mortality in Finnish men

Aino Siltari [1,2]☯ *, Teemu J. Murtola[1,3]☯, Kirsi Talala[4], Kimmo Taari[5], Teuvo L. J. Tammela[1,3], Anssi Auvinen[6]

1 Faculty of Medicine and Health Technology, Tampere University, Tampere, Finland, 2 Department of Pharmacology, Faculty of Medicine, University of Helsinki, Helsinki, Finland, 3 Department of Urology, TAYS Cancer Center, Tampere, Finland, 4 Finnish Cancer Registry, Helsinki, Finland, 5 Department of Urology, University of Helsinki and Helsinki University Hospital, Helsinki, Finland, 6 School of Health Sciences, Tampere University, Tampere, Finland

☯ These authors contributed equally to this work.
* aino.siltari@helsinki.fi

**Data Availability Statement:** Sharing individual-level data, even in pseudonymized form is not possible according to the current Finnish regulation regarding privacy, data protection and EU-level GDPR. Full anonymization of the data is not

## Abstract

The aim of this study was to investigate pre- and post-diagnostic use of antihypertensive drugs on prostate cancer (PCa)-specific survival and the initiation of androgen deprivation therapy (ADT). The cohort investigated 8,253 PCa patients with 837 PCa-specific deaths during the median follow-up of 7.6 years after diagnosis. Information on drug use, cancer incidence, clinical features of PCa, and causes of death was collected from Finnish registries. Hazard ratios with 95% confidence intervals were calculated using Cox regression with antihypertensive drug use as a time-dependent variable. Separate analyses were performed on PCa survival related to pre- and post-diagnostic use of drugs and on the initiation of ADT. Antihypertensive drug use overall was associated with an increased risk of PCa-specific death (Pre-PCa: 1.21 (1.04–1.4), Post-PCa: 1.2 (1.02–1.41)). With respect to the separate drug groups, angiotensin II type 1 receptor (ATr) blockers, were associated with improved survival (Post-PCa: 0.81 (0.67–0.99)) and diuretics with an increased risk (Post-PCa: 1.25 (1.05–1.49)). The risk of ADT initiation was slightly higher among antihypertensive drug users as compared to non-users. In conclusion, this study supports anti-cancer effect of ATr blockers on PCa prognosis and this should be investigated further in controlled clinical trials.

## Introduction

Prostate cancer (PCa) is the most common cancer in men [1]. Known risk factors for PCa are age, race, and a family history of PCa. Hypertension has been suggested as a risk factor for PCa progression [2–4]. However, hypertension is linked to other factors such as metabolic syndrome—thus, it is difficult to distinguish the impacts of underlying risk factors, hypertension and its medication on PCa development and progression. As in many other cancers, PCa involves re-programmed normal cellular functions, such as glucose- and cholesterol metabolism, in cancer progression [5]. Thus, commonly used drugs affecting normal cellular

feasible, because at individual level, there would be numerous unique records if all variables used in the analyses were to be retained (even if collapsing discrete variables and categorizing continuous variables). Permission for data access was applied from Finnish institute of health and welfare and data access can be applied from there (kirjaamo@thl.fi).

**Funding:** This work was supported by Pirkanmaa Hospital District in the form of a grant awarded to TM (9T036), Cancer Foundation Finland in the form of a grant awarded to TM (25024194), and Finnish-Norwegian Medical Foundation in the form of a grant awarded to AS (201900067). The funders had no role in study design, data collection and analysis, decision to publish, or preparation of the manuscript.

**Competing interests:** The authors have declared that no competing interests exist.

functions such as drugs targeting the renin-angiotensin system (RAS), fluid homeostasis, and the sympathetic nervous system may influence PCa progression.

In particular, the role of RAS in cancer development and progression has been under investigation [6, 7]. In mice, treatment with angiotensin II receptor blockers decreased the volume of a prostate tumor. However, administration of angiotensin II had no effect on tumor size [8]. On the other hand, angiotensin II has increased prostate cell viability in different cell models [9, 10].

Results on antihypertensive drug use and PCa-specific mortality are controversial [11–18]. In general, the evaluation of use of different antihypertensive drug groups is challenging due to the heterogeneity of drug users. Furthermore, differentially acting drug groups are commonly used in parallel to achieve blood pressure control.

Recently, we showed that the use of antihypertensive drugs is moderately associated with an increased risk for prostate cancer in a comprehensive population-based cohort study based on the Finnish Randomized Study of Screening for Prostate Cancer (FinRSPC). The risk increase was not related to any specific drug group [19]. Here we have investigated whether pre- and post-diagnostic use of any antihypertensive drug group associates with PCa-specific survival. Furthermore, we evaluated the risk of PCa progression by using initiation of androgen-deprivation therapy (ADT) as a surrogate.

## Materials and methods

### Study cohort

The original study cohort, FinRSPC, involved 80,458 men aged 55–67 years old at study entry from Tampere and Helsinki areas in Finland [20]. All men were free of PCa at baseline. By the end of 2016, 8,253 men were diagnosed with prostate cancer and information was available on their drug use. These men formed our study population. The Finnish Cancer Registry, which covers 96% of all solid tumor cases, provided information on cancer diagnoses [21]. Data on Gleason scores, TNM stage, and prostate cancer-specific antigen (PSA) level at the time of diagnosis was obtained from medical records. The Cause of Death registry in Statistics Finland houses information on causes of death (TK-53-1330-18) and the Care Register for Health Care provided information on diagnoses and medical procedures in secondary and tertiary health care units. Unique personal identification number, assigned to all Finnish citizens, was used in a deterministic linkage to combine information from the registries and medical records. Causes of death were classified using the International Classification of Diseases (ICD-10) codes; PCa was considered the cause of death when the underlying cause was C61 and deaths from cardiovascular disease (CVD) included codes I20-I25, I30-I52, and I70-I79. During 2004–2008, a questionnaire about height, weight, and use of non-prescription drugs, such aspirin and other non-steroidal anti-inflammatory drugs (NSAIDs) was mailed to participants still in the study. Originally, information on Body Mass Index (BMI) was collected from 11,698 subjects [22], and in the current study population, this was available for 805 men.

### Information on medication use

The prescription database of Social Insurance Institution (SII) of Finland was used to collect information on the use of antihypertensive drugs from 1996 to 2016. As a part of the national health insurance, SII provides reimbursements for purchases of physician-prescribed drugs to all Finnish citizens. All reimbursements are recorded in the database including the purchase date, drug dose, number of doses in the package, and number of drug packages for each purchase. In Finland, all antihypertensive drugs are available only by prescription, thus all purchases, except drugs for hospital patients, are recorded.

Drug-specific Anatomic Therapeutic Chemical (ATC) codes were used to identify the medications and were divided according to mechanism of action into angiotensin-converting enzyme (ACE) inhibitors, angiotensin II receptor (ATr) blockers, beta-blockers, calcium channel blockers, and diuretics. Additional information on purchases of statins, antidiabetic drugs, 5-alpha-reductase inhibitors, and prescription aspirin and other NSAIDs was also collected. The diuretics analyzed included only those drugs used for hypertension. Thus, loop-diuretics and spironolactone, commonly used for edema and fluid retention problems, were excluded.

Defined Daily Dose (DDD) [23] was used to calculate standardized cumulative doses by dividing annual amount of purchased drug by drug-specific DDD-values. All DDDs in each group were calculated together and value presented as annual cumulative doses of each drug group. The number of years with recorded drug purchases was used to calculate cumulative years of drug use. By dividing cumulative doses by cumulative years, average annual doses, i.e. average intensities of the drug use were calculated. Risk trends were analyzed by stratifying the study population by tertiles of intensity, DDD-values, and years of the drug use (termed low, medium and high use).

## Statistical analysis

Hazard ratios (HR) and their 95% confidence intervals (CI) for prostate cancer-specific mortality were estimated using Cox regression models. The time metric was months and years from diagnosis. Each antihypertensive drug group was analyzed as a time-dependent variable, except pre-diagnostic use which was analyzed as a time-fixed variable (thus subjects were classified as either ever-users or never-users before the diagnosis). Purchases were used to determine annually cumulative use for each follow-up year for each drug group. Subjects stayed as non-users until the first antihypertensive drug purchase and after that, they remained as ever-users for the whole follow-up period. This minimized bias due to selective discontinuation of medication in the terminal phase of cancer. Analysis of overall antihypertensive drug use was conducted separately. Different drug groups were included into models simultaneously as separate time-dependent variables, to enable modelling of mutually adjusted simultaneous use of several antihypertensive drugs. Lag-time analysis was conducted by lagging the diagnosis of PCa by one or three years from different drug use.

To evaluate whether antihypertensive drug use exerted any impact on the progression of cancer, the risk for initiation of androgen deprivation therapy (ADT) was analyzed using a Cox regression model, where follow-up started at PCa diagnosis and continued until ADT initiation, death/emigration, or the end of 2016. The analysis was limited only for long-term ADT treatment, participants who also had radiation therapy as primary management in addition to ADT were excluded.

Analyses were performed for risk overall and separately for subgroups of Gleason grade 7 and 8–10, risk group 2 (for definition see below), and metastatic cancers. The risk for all-cause mortality was calculated without subgroup analyses. All analyses were adjusted for age, FinRSPC trial arm (screening arm and control arm), year of diagnosis, cancer clinical characteristics (T stage, metastasis, and Gleason grade), Charlson comorbidity index, use of statins, antidiabetic drugs, anticoagulants, 5-alpha-reductase inhibitors, aspirin and other NSAIDs. Sensitivity analysis by adjusting the model by marital and socioeconomical status, and BMI were conducted with subgroup in whom information was available. We also conducted sensitivity analysis by number of used drugs (one, two, or three or more) and combination of different drug groups (e.g. ATr blockers + diuretics). PCa risk groups were created based on Gleason grade, clinical characteristics and tumor extent: risk group 0 included cases where Gleason grade was 6, T-stage was 1 or 2, or PSA was less than 10 µg/l, risk group 1 included

cases where Gleason grade was 7, T-stage was 3, or PSA was 10–20 μg/l and risk group 2 included cases where Gleason grade was 8 or more, T-stage was 4, cancer was metastatic, or PSA was more than 20 μg/l. Charlson comorbidity index was calculated as explained previously [24]. All analyses were performed using IBM SPSS statistical software (version 24).

We performed competing risks analysis to estimate association between antihypertensive drug groups and PCa-specific death, with deaths from cardiovascular disease (CVD) as the competing risk. Deaths due to non-cancer and non-CVD causes were censored. The model was adjusted for age, FinRSPC trial arm, year of diagnosis, PCa risk group, Charlson comorbidity index, use of statins, antidiabetic drugs, anticoagulants, 5-alpha-reductase inhibitors, aspirin, and other NSAIDs. The analysis was done using StataCorp Stata Statistics (version 14.0).

## Results

### Population characteristics

The majority (79%) of the study population had at least one antihypertensive drug purchase during the follow-up (Table 1). Of the 8,253 prostate cancer cases, 2,479 had Gleason grade 7 and 1,379 Gleason 8 or above, 2284 patients belonged to risk group 2, and 589 had metastatic cancer. The median follow-up time was 7.6 years after diagnosis. The median age at diagnosis was 68 years. In total, 2,765 subjects died during the follow-up, including 837 deaths from PCa (Table 1). The overall PCa-specific death rate after diagnosis was 13.3 per 1000 person-years. It was 12.0 among users and 20.2 in non-users (Table 1). Of the specific drug groups, PCa-specific mortality after diagnosis was highest among users of diuretics (11.7 deaths per 1000 person-years) and lowest among users of ATr blockers (8 per 1000) (Table 1).

The users of antihypertensive drugs also had been prescribed other drugs more often than non-users (p<0.001) (Table 1). Non-users had slightly lower BMI values. There was no difference in the distribution of FinRSPC study arms or age at diagnosis between the groups (68 or 69 years in all groups). The follow-up time was slightly shorter in the group of non-users, however, it was similar between all other groups (Table 1).

### Risk of prostate cancer-specific death by antihypertensive drug use

Both pre- and post-diagnostic use of antihypertensive drugs was associated with an increased risk for PCa-specific death compared to non-users in multivariable-adjusted analysis (HR 1.21, 95% CI 1.04–1.4 and HR 1.2, 95% CI 1.02–1.41, respectively) (Fig 1, Table 2). In addition, the risk for all-cause mortality was higher among users of antihypertensive drugs (Pre: HR 1.38, 95% CI 1.27–1.5, Post: HR 1.33, 95% CI 1.21–1.47) (Table 3A). As expected, the risk of CVD-specific deaths was elevated in users of antihypertensive drugs (Pre: HR 1.96 95% CI 1.61–2.39, Post: HR 2.57, 95% CI 1.97–3.36) (Table 3B).

When antihypertensive drug groups were analyzed separately, post-diagnostic use of ATr blockers was associated with a decreased risk for PCa death (HR 0.81, 95% CI 0.67–0.99) (Fig 1, Table 2). The risk decrease was observed also for pre-diagnostic use (HR 0.74, 95% CI 0.58–0.96). Pre- and post-diagnostic use of diuretics was associated with an increased risk of death from PCa (Pre: HR 1.31, 95% CI 1.07–1.6, Post: HR 1.25, 95% CI 1.05–1.49) (Fig 1, Table 2). Tumor characteristics did not modify the risk associations by the use of ATr blockers or diuretics (Fig 2, Table 2).

Furthermore, pre-diagnostic use of ATr blockers was associated with a decreased all-cause mortality (Table 3A). On the other hand, the use of beta-blockers and ACE inhibitors was associated with increased all-cause mortality (Table 3A). Other investigated antihypertensive drug group did not associate with the risk of PCa-specific or all-cause mortality (Fig 1,

**Table 1. Population characteristics.** Cohort of 8253 men with prostate cancer (PCa) from the Finnish Randomized Study of Screening for Prostate Cancer. ACE inhibitors = angiotensin-converting enzyme inhibitors; ATr blockers = angiotensin II receptor type.

| | All | Non-users | Users | ACE inbitors | ATr blockers | Beta-blockers | Calcium channel blockers | Diuretics |
|---|---|---|---|---|---|---|---|---|
| NO. of PCa cases | 8253 | 1875 (22.7) | 6378 (77.3) | 3297 (39.9) | 2316 (28.1) | 4467 (54.1) | 3172 (38.4) | 2397 (29) |
| Gleason 7 n (%) | 2479 (30) | 539 (28.7) | 1940 (30.5) | 990 (30.0) | 694 (30.0) | 1355 (30.3) | 958 (30.2) | 705 (29.4) |
| Gleason 8–10 n (%) | 1379 (16.7) | 345 (18.4) | 1034 (16.2) | 510 (15.5) | 352 (15.2) | 705 (15.8) | 464 (14.6) | 361 (15.1) |
| Risk group 2 n (%) | 2284 (27.7) | 559 (29.8) | 1725 (27) | 877 (26.6) | 560 (24.2) | 1211 (27.1) | 811 (25.6) | 627 (26.2) |
| Metastatic ceses n (%) | 589 (7.1) | 180 (9.6) | 409 (6.4) | 208 (6.3) | 99 (4.3) | 277 (6.2) | 173 (5.5) | 141 (5.9) |
| Hormonal therapy n (%) (excluded cases where radiation therapy was combined with ADT) | 2120 (25.7) | 483 (25.8) | 1637 (25.7) | 872 (26.4) | 469 (20.3) | 1149 (25.7) | 764 (24.1) | 581 (24.2) |
| Median age at time of randomization (IQR) | 63 (59–63) | 59 (55–63) | 63 (59–67) | 63 (59–67) | 59 (59–63) | 63 (59–67) | 63 (59–67) | 63 (59–67) |
| Median age at time of diagnosis (IQR) | 68 (64–72) | 68 (64–72) | 69 (65–72) | 68 (64–72) | 68 (64–72) | 69 (65–73) | 68 (65–72) | 68 (64–72) |
| Follow-up time after randomization, median (IQR) | 12 (11–13) | 12 (11–13) | 12 (11–13) | 12 (11–13) | 12 (11–13.4) | 12 (11–13) | 12 (11–13) | 12 (11–13) |
| Follow-up time after diagnosis, median (IQR) | 7.6 (3.8–11.1) | 5.9 (2.8–10.3) | 8 (4.2–11.3) | 8.4 (4.6–11.8) | 8.7 (4.8–11.8) | 8.1 (4.3–11.5) | 8.5 (4.7–11.7) | 8.9 (5–12) |
| Deaths n (%) | 2765 (33.5) | 622 (33.2) | 2143 (33.6) | 1148 (34.8) | 663 (28.6) | 1587 (35.5) | 1042 (32.8) | 805 (33.6) |
| PCa-specifin deaths n (% from all deaths) | 837 (30.2) | 224 (36) | 613 (28.6) | 308 (26.8) | 162 (24.4) | 415 (26.1) | 271 (26) | 249 (30.9) |
| PCa-specifin deaths per 1000 person-years after diagnosis | 13.3 | 20.2 | 12.0 | 11.1 | 8.0 | 11.5 | 10.1 | 11.7 |
| CVD-specific deaths n (% from all deaths) | 586 (21.2) | 66 (10.6) | 520 (8.2) | 328 (28.6) | 175 (26.4) | 420 (26.5) | 259 (24.9) | 197 (24.4) |
| Median BMI (IQR) (n) | 26.24 (24.22–28.67) (805) | 25.06 (23.54–27.17) (197) | 26.59 (24.55–28.73) (608)*** | 26.6 (24.4–29.0) (277)*** | 27.1 (25.0–29.1) (242)*** | 27.0 (24.9–29.1) (391)*** | 26.9 (24.9–29.1) (299)*** | 27.74 (25.42–30.05) (202) |
| **FinRSPC arm** | | | | | | | | |
| Screening n (%) | 3425 (41.5) | 797 (42.5) | 2628 (41.2) | 1391 (42.2) | 934 (40.3) | 1855 (41.5) | 1318 (41.6) | 962 (40.1) |
| Control n (%) | 4828 (58.5) | 1078 (57.5) | 3750 (58.8) | 1906 (57.8) | 1382 (59.7) | 2612 (58.5) | 1854 (58.4) | 1435 (59.9) |
| **Use of other drugs** | | | | | | | | |
| Statins n (%) | 3870 (46.9) | 408 (21.8) | 3462 (54.3)*** | 1913 (58.0)*** | 1383 (59.7)*** | 2604 (58.3)*** | 1825 (57.5)*** | 1382 (57.7)*** |
| Antidiabetic drugs n (%) | 1599 (19.4) | 118 (6.3) | 1481 (23.3)*** | 915 (27.8)*** | 608 (26.2)*** | 1130 (25.3)*** | 853 (26.9)*** | 617 (25.7)*** |
| NSAIDs n (%) | 7065 (85.6) | 1534 (81.8) | 5531 (86.7)*** | 2821 (85.6)*** | 2058 (88.9)*** | 3881 (86.9)*** | 2771 (87.4)*** | 2107 (87.9)*** |
| Aspirin n (%) | 1143 (13.8) | 139 (7.4) | 1004 (15.7)*** | 522 (15.8)*** | 404 (17.4)*** | 760 (17.0)*** | 528 (16.6)*** | 374 (15.6)*** |
| 5-alpha-reductase inhibitors n (%) | 1229 (14.9) | 224 (11.9) | 1005 (15.8)*** | 495 (15.0)*** | 381 (16.5)*** | 717 (16.1)*** | 499 (15.7)*** | 390 (16.3)*** |
| Anticoagulants n (%) | 4275 (51.8) | 585 (31.2) | 3690 (57.9)*** | 2050 (62.2)*** | 1370 (59.2)*** | 2895 (64.8)*** | 1867 (58.9)*** | 1388 (57.9)*** |

Tables 2 & 3A). The use of beta-blockers and ACE inhibitors also associated with elevated CVD mortality (Table 3B). Diuretics and calcium channel blockers displayed no association with deaths from CVD.

Adjustments of the model by socioeconomical and marital status, and BMI did not modify the risk associations. Lagging the PCa diagnosis by one or three years after drug use did not reveal any statistically meaningful association with prostate cancer-specific mortality (S3 Table).

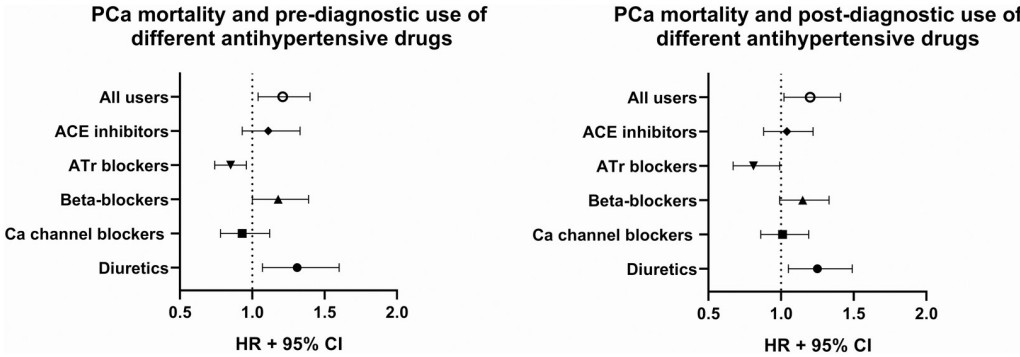

**Fig 1. Association of prostate cancer (PCa)-specific mortality and pre- and post-diagnostic use of different antihypertensive drugs.** Data is presented as hazard ratios (HR) with 95% confidence intervals (CI).

Sensitivity analysis by number of used drugs did not modify the results as risk for PCa-specific death was increased similarly despite number of drug groups in use (S5 Table). Some drug combinations showed slightly different survival associations compared to main analysis; combination of beta-blockers and diuretics were associated with lowered risk of PCa death compared non-users (HR 0.74, 95% CI 0.57–0.96) whereas diuretics and beta-blockers in combination were associated with increased risk (HR 2.36, 95% CI 1.89–2.94). Other drug group combination did not show significant associations with PCa-specific survival (S5 Table).

### Risk trends by cumulative antihypertensive drug use

In the risk trend analysis for the post-diagnostic use of ACE inhibitors, the risk for PCa death tended to decrease with the amount of use, but the trend was not statistically significant (Table 4). Among ATr blocker users, the strongest risk decrease was observed in the tertile of highest intensity of use (over 559.2 DDDs annually) regardless of cancer grade and stage (Table 4). No clear dose-dependent risk trends were observed by pre-diagnostic use for either drug group (S1 Table). Risk trend analyses for tertiles of post-diagnostic DDD-values and years of use can be found in the supplemental material (S2 Table).

### Risk for initiation of androgen deprivation therapy

Any use of antihypertensive drugs was associated with a slightly increased risk for initiation of ADT as compared to non-users (HR 1.15, 95% CI 1.05–1.27) (Table 4). The risk increase was seen also in Gleason 8–10 and risk group 2 tumors (Table 5). When different drug groups were compared, only use of ACE inhibitors and beta blockers showed increased risks (HR 1.21, 95% CI 1.09–1.35, and HR 1.1, 95% CI 1–1.21, respectively). Such associations were not seen in the subgroups of clinical tumor characteristics (Table 5). Drug groups did not reveal any meaningful dose-dependent association with the risk of initiation of ADT (S4 Table).

### Competing risk analysis

The risk association for the pre-diagnostic use antihypertensive drug use was similar in competing risk analysis as in the main analysis; use of ATr blockers continued to be associated with improved disease-specific survival (HR 0.74, 95% CI 0.57–0.98) and the use of diuretics slightly increased the risk (HR 1.25, 95% CI 1.01–1.54).

**Table 2. Risk for prostate cancer (PCa)-specific mortality of pre- and post-diagnostic use of antihypertensive drugs compared to non-users after PCa diagnosis.**
Cox regression hazard model was adjusted with age, FinRSPC trial arm (screening arm and control arm), year of diagnosis, cancer clinical characteristics (T stage, metastasis, and Gleason grade), Charlson comorbidity index, use of statins, antidiabetic drugs, anticoagulants, 5-alpha-reductase inhibitors, aspirin and other NSAIDs. ACE inhibitors = angiotensin-converting enzyme inhibitors; ATr blockers = angiotensin II receptor type 1 blockers; HR (95% CI) = hazard ratio and 95% confidence intervals.

| PCa-specific death | | | Pre-diagnostic use | Post-diagnostic use |
|---|---|---|---|---|
| Overall risk | n of men | n of PCa deaths (%) | HR (95% CI) | HR (95% CI) |
| Non-users | 1875 | 224 (11.9) | ref. | ref. |
| Users | 6378 | 613 (9.6) | 1.21 (1.04–1.4) | 1.2 (1.02–1.41) |
| ACE inhibitors | 3297 | 308 (9.3) | 1.11 (0.93–1.33) | 1.04 (0.88–1.22) |
| ATr blockers | 2316 | 162 (7) | 0.74 (0.58–0.96) | 0.81 (0.67–0.99) |
| Beta-blockers | 4467 | 415 (9.3) | 1.18 (1.0–1.39) | 1.15 (0.99–1.33) |
| Calcium channel blockers | 3172 | 271 (8.5) | 0.93 (0.78–1.12) | 1.01 (0.86–1.19) |
| Diuretics | 2397 | 249 (10.4) | 1.31 (1.07–1.6) | 1.25 (1.05–1.49) |
| **Gleason 7** | | | | |
| Non-users | 539 | 50 | ref. | ref. |
| Users | 1940 | 178 | 1.69 (1.28–2.26) | 1.39 (1–1.94) |
| ACE inhibitors | 1489 | 130 | 1.26 (0.89–1.78) | 1.26 (0.97–1.76) |
| ATr blockers | 1785 | 179 | 0.89 (0.56–1.43) | 0.76 (0.53–1.09) |
| Beta-blockers | 1124 | 111 | 1.31 (0.96–1.8) | 1.07 (0.79–1.44) |
| Calcium channel blockers | 1521 | 141 | 0.86 (0.6–1.23) | 1.05 (0.78–1.41) |
| Diuretics | 1774 | 145 | 1.48 (1.1–2.16) | 1.55 (1.13–2.15) |
| **Gleason 8–10** | | | | |
| Non-users | 345 | 115 | ref. | ref. |
| Users | 1034 | 288 | 1.05 (0.84–1.31) | 1.11 (0.88–1.41) |
| ACE inhibitors | 510 | 134 | 1.1 (0.85–1.42) | 0.91 (0.72–1.15) |
| ATr blockers | 352 | 81 | 0.72 (0.51–1.02) | 0.9 (0.68–1.2) |
| Beta-blockers | 705 | 195 | 1.14 (0.9–1.45) | 1.12 (0.96–1.48) |
| Calcium channel blockers | 464 | 119 | 0.98 (0.76–1.26) | 1.01 (0.8–1.28) |
| Diuretics | 361 | 103 | 1.24 (0.94–1.65) | 1.08 (0.8–1.28) |
| **Risk group 2** | | | | |
| Non-users | 559 | 179 | ref. | ref. |
| Users | 1725 | 426 | 1.12 (0.94–1.33) | 1.1 (0.91–1.33) |
| ACE inhibitors | 1407 | 214 | 1.18 (0.95–1.45) | 1.1 (0.91–1.33) |
| ATr blockers | 1724 | 106 | 0.7 (0.52–0.94) | 0.82 (0.65–1.05) |
| Beta-blockers | 1073 | 282 | 1.13 (0.94–1.38) | 1.08 (0.9–1.23) |
| Calcium channel blockers | 1473 | 181 | 0.93 (0.76–1.15) | 0.99 (0.82–1.2) |
| Diuretics | 1657 | 163 | 1.31 (1.03–1.66) | 1.15 (0.93–1.42) |
| **Metastatic disease** | | | | |
| Non-users | 180 | 112 | ref. | ref. |
| Users | 409 | 213 | 1.04 (0.82–1.32) | 0.95 (0.74–1.22) |
| ACE inhibitors | 381 | 106 | 1.38 (1.02–1.85) | 1.1 (0.83–1.45) |
| ATr blockers | 490 | 41 | 0.73 (0.48–1.12) | 0.83 (0.57–1.22) |
| Beta-blockers | 312 | 131 | 0.95 (0.73–1.24) | 0.93 (0.73–1.19) |
| Calcium channel blockers | 416 | 87 | 1 (0.75–1.34) | 1.08 (0.82–1.42) |
| Diuretics | 448 | 67 | 1.01 (0.73–1.41) | 0.89 (0.64–1.23) |

## Discussion

We evaluated prostate cancer-specific and all-cause mortality by pre- and post-diagnostic anti-hypertensive drug use in a Finnish cohort study consisting of men living in metropolitan areas

**Table 3. Risk for all-cause mortality (A), and cardiovascular diseases (CVD) mortality of pre- and post-diagnostic use of antihypertensive drugs compared to non-users after PCa diagnosis.** Cox regression hazard model was adjusted with age, FinRSPC trial arm (screening arm and control arm), year of diagnosis, cancer clinical characteristics (T stage, metastasis, and Gleason grade), Charlson comorbidity index, use of statins, antidiabetic drugs, anticoagulants, 5-alpha-reductase inhibitors, aspirin and other NSAIDs. ACE inhibitors = angiotensin-converting enzyme inhibitors; ATr blockers = angiotensin II receptor type 1 blockers; HR (95% CI) = hazard ratio and 95% confidence intervals.

| A) | | | | |
|---|---|---|---|---|
| **All-cause death** | | | **Pre-diagnostic use** | **Post-diagnostic use** |
| **Overall risk** | n of men | n of deaths | HR (95% CI) | HR (95% CI) |
| Non-users | 1875 | 622 | ref. | ref. |
| Users | 6378 | 2143 | 1.38 (1.27–1.5) | 1.33 (1.21–1.47) |
| ACE inhibitors | 3297 | 1148 | 1.22 (1.11–1.34) | 1.19 (1.09–1.29) |
| ATr blockers | 2316 | 663 | 0.88 (0.77–1) | 0.98 (0.89–1.08) |
| Beta-blockers | 4467 | 1587 | 1.22 (1.12–1.33) | 1.4 (1.28–1.52) |
| Calcium channel blockers | 3172 | 1024 | 1.05 (0.96–1.16) | 1.07 (0.99–1.17) |
| Diuretics | 2397 | 805 | 1.16 (1.05–1.29) | 0.97 (0.88–1.07) |
| B) | | | | |
| **CVD death** | | | **Pre-diagnostic use** | **Post-diagnostic use** |
| **Overall risk** | n of men | n of CVD deaths | HR (95% CI) | HR (95% CI) |
| Non-users | 1875 | 66 | ref. | ref. |
| Users | 6378 | 520 | 1.96 (1.61–2.39) | 2.57 (1.97–3.36) |
| ACE inhibitors | 3297 | 328 | 1.52 (1.25–1.85) | 1.85 (1.54–2.22) |
| ATr blockers | 2316 | 175 | 0.83 (0.64–1.08) | 1.26 (1.03–1.54) |
| Beta-blockers | 4467 | 420 | 1.67 (1.39–2.02) | 2.07 (1.7–2.53) |
| Calcium channel blockers | 3172 | 259 | 1.12 (0.92–1.36) | 0.88 (0.72–1.06) |
| Diuretics | 2397 | 197 | 1.11 (0.9–1.37) | 1.13 (0.95–1.36) |

of Helsinki and Tampere. In general, the use of antihypertensive drugs was associated with increased PCa-specific and all-cause mortality as compared to non-users. When we evaluated the different antihypertensive drug groups, post-diagnostic use of RAS-inhibiting drugs, ACE inhibitors, and AT-receptor blockers were associated with improved survival, whereas diuretics were associated with poorer survival.

The reduction in risk was more pronounced for AT-receptor blockers than for ACE inhibitors. However, only post-diagnostic use of ACE inhibitors showed a dose-dependent risk trend. Similarly, in a UK population-based cohort, Cardwell et al. [13] revealed that users of ACE inhibitors and ATr blockers had a slightly decreased PCa-specific mortality and concluded that it was safe to use these antihypertensive drugs after PCa diagnosis. Furthermore, Ronquist et al. [25] observed that the post-operative captopril users had less biochemical

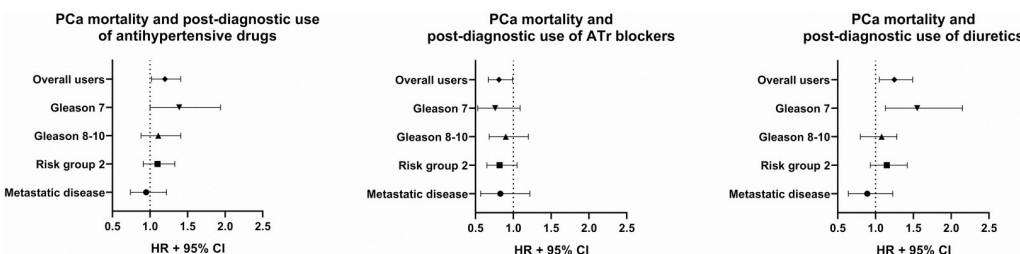

**Fig 2. Role of tumor clinical characteristics on PCa mortality among users of any antihypertensive drug, and separately for angiotensin II receptor blockers (ATr blockers) and diuretics.** Data is presented as hazard ratios (HR) with 95% confidence intervals (CI).

**Table 4. Risk of prostate cancer (PCa)-specific mortality by the post-diagnostic use of antihypertensive drugs after diagnosis of PCa.** Users stratified into tertiles by cumulative intensity of the use (DDD values/years of the use). Cox regression model was adjusted with age, FinRSPC trial arm (screening arm and control arm), year of diagnosis, cancer clinical characteristics (T stage, metastasis, and Gleason grade), Charlson comorbidity index, use of statins, antidiabetic drugs, anticoagulants, 5-alpha-reductase inhibitors, aspirin and other NSAIDs. ACE inhibitors = angiotensin-converting enzyme inhibitors; ATr blockers = angiotensin II receptor type 1 blockers; HR (95% CI) = hazard ratio with 95% confidence intervals.

| | ACE inhibitors | ATr blockers | Beta-blockers | Calcium channel blockers | Diuretics |
|---|---|---|---|---|---|
| **Limits** | | | | | |
| Low | <336 | <303.2 | <100 | <261.3 | <148.3 |
| Medium | 336–676.9 | 303.2–559.2 | 100–187.4 | 261.3–399.4 | 148.3–217.8 |
| High | 676.9> | 559.2> | 187.4> | 399.4> | 217.8> |
| **Overall, n** | | | | | |
| Low | 1100 | 772 | 1542 | 1051 | 790 |
| Medium | 1098 | 772 | 1436 | 1064 | 790 |
| High | 1099 | 772 | 1489 | 1057 | 790 |
| **Overall, PCa death risk** | HR (95% CI) | HR (95% CI) | HR (95% CI) | HR (95% CI) | HR (95% CI) |
| Low | 1.16 (0.93–1.45) | 0.89 (0.67–1.18) | 1.27 (1.04–1.55) | 1.18 (0.94–1.48) | 1.44 (1.12–1.84) |
| Medium | 1.06 (0.84–1.33) | 0.81 (0.6–1.09) | 1.29 (1.05–1.6) | 0.93 (0.73–1.19) | 1.09 (0.82–1.45) |
| High | 0.92 (0.72–1.19) | 0.63 (0.45–0.88) | 0.97 (0.78–1.21) | 1.08 (0.84–1.4) | 1.32 (1.02–1.7) |
| **Gleason 7, n** | | | | | |
| Low | 302 | 201 | 465 | 379 | 240 |
| Medium | 337 | 255 | 228 | 408 | 215 |
| High | 351 | 238 | 462 | 315 | 241 |
| **Gleason 7, PCa death risk** | HR (95% CI) | HR (95% CI) | HR (95% CI) | HR (95% CI) | HR (95% CI) |
| Low | 1.15 (0.75–1.78) | 0.99 (0.6–1.64) | 1.04 (0.71–1.55) | 1.11 (0.74–1.65) | 1.92 (1.27–2.91) |
| Medium | 1.28 (0.84–1.96) | 0.76 (0.44–1.33) | 0.92 (0.6–1.42) | 1.15 (0.75–1.75) | 1.3 (0.78–2.15) |
| High | 1.67 (1.09–2.57) | 0.69 (0.37–1.26) | 1.17 (0.78–1.74) | 0.92 (0.55–1.55) | 1.28 (0.79–2.08) |
| **Gleason 8–10, n** | | | | | |
| Low | 168 | 134 | 256 | 288 | 104 |
| Mediate | 178 | 102 | 229 | 226 | 132 |
| High | 164 | 116 | 220 | 116 | 119 |
| **Gleason 8–10, PCa death risk** | HR (95% CI) | HR (95% CI) | HR (95% CI) | HR (95% CI) | HR (95% CI) |
| Low | 1.06 (0.75–1.48) | 0.94 (0.63–1.41) | 1.58 (1.18–2.12) | 1.38 (0.97–1.97) | 1.21 (0.81–1.81) |
| Medium | 0.89 (0.63–1.25) | 0.85 (0.55–1.31) | 1.5 (1.11–2.02) | 0.95 (0.65–1.38) | 1.02 (0.67–1.55) |
| High | 0.69 (0.47–1.02) | 0.58 (0.36–0.95) | 0.85 (0.61–1.2) | 1.15 (0.81–1.65) | 1.24 (0.84–1.83) |
| **Risk group 2, n** | | | | | |
| Low | 287 | 201 | 417 | 476 | 207 |
| Medium | 301 | 183 | 394 | 369 | 220 |
| High | 289 | 176 | 400 | 231 | 193 |
| **Risk group 2, PCa death risk** | HR (95% CI) | HR (95% CI) | HR (95% CI) | HR (95% CI) | HR (95% CI) |
| Low | 1.23 (0.94–1.61) | 0.9 (0.64–1.27) | 1.28 (1–1.62) | 1.17 (0.89–1.54) | 1.27 (0.93–1.74) |
| Medium | 1.06 (0.81–1.41) | 0.77 (0.53–1.13) | 1.25 (0.97–1.62) | 0.87 (0.64–1.18) | 1.09 (0.78–1.53) |
| High | 1.01 (0.75–1.35) | 0.62 (0.41–0.94) | 0.91 (0.7–1.19) | 1.15 (0.85–1.55) | 1.25 (0.91–1.71) |
| **Metastatic cancer, n** | | | | | |
| Low | 77 | 33 | 104 | 154 | 43 |
| Medium | 69 | 39 | 91 | 78 | 47 |
| High | 62 | 27 | 82 | 27 | 48 |
| **Metastatic cancer, PCa death risk** | HR (95% CI) | HR (95% CI) | HR (95% CI) | HR (95% CI) | HR (95% CI) |
| Low | 1.26 (0.86–1.85) | 0.91 (0.52–1.61) | 1.02 (0.72–1.44) | 1.37 (0.93–2.03) | 0.79 (0.47–1.33) |
| Medium | 1.24 (0.84–1.83) | 0.9 (0.52–1.56) | 1.4 (0.98–2) | 0.75 (0.46–1.21) | 0.83 (0.5–1.38) |
| High | 1.02 (0.67–1.55) | 0.42 (0.2–0.88) | 0.62 (0.42–0.91) | 1.55 (1.02–2.33) | 1.29 (0.81–2.07) |

**Table 5. Effect of antihypertensive drugs on initiation of hormonal therapy for treatment of prostate cancer in Finnish men.** Cox regression hazard model was adjusted with age, FinRSPC trial arm (screening arm and control arm), year of diagnosis, cancer clinical characteristics (T stage, metastasis, and Gleason grade), Charlson comorbidity index, use of statins, antidiabetic drugs, anticoagulants, 5-alpha-reductase inhibitors, aspirin and other NSAIDs. ADT = androgen deprivation therapy; ACE inhibitors = angiotensin-converting enzyme inhibitors; ATr blockers = angiotensin II type 1 receptor blockers; HR (95% CI) = hazard ratio with 95% confidence intervals.

| | | Initiation of ADT |
|---|---|---|
| **Overall risk** | **n of ADT treated men** | **HR (95% CI)** |
| Non-users | 483 | ref. |
| Users | 1637 | 1.15 (1.05–1.27) |
| ACE inhibitors | 872 | 1.21 (1.09–1.35) |
| ATr blockers | 469 | 1.05 (0.91–1.21) |
| Beta-blockers | 1149 | 1.1 (1-1-21) |
| Calcium channel blockers | 764 | 0.86 (0.76–0.98) |
| Diuretics | 581 | 1 (0.89–1.11) |
| **Gleason 7** | | |
| Non-users | 298 | ref. |
| Users | 958 | 1.13 (0.94–1.36) |
| ACE inhibitors | 270 | 1.16 (0.95–1.41) |
| ATr blockers | 141 | 1.07 (0.83–1.39) |
| Beta-blockers | 356 | 1.12 (0.94–1.34) |
| Calcium channel blockers | 243 | 1.05 (0.86–1.29) |
| Diuretics | 335 | 0.8 (0.62–1.02) |
| **Gleason 8–10** | | |
| Non-users | 219 | ref. |
| Users | 548 | 1.2 (1–1.43) |
| ACE inhibitors | 231 | 1.22 (0.99–1.5) |
| ATr blockers | 129 | 1.11 (0.84–1.46) |
| Beta-blockers | 319 | 1.09 (0.91–1.31) |
| Calcium channel blockers | 205 | 099 (0.81–1.23) |
| Diuretics | 184 | 0.92 (0.81–1.23) |
| **Risk group 2** | | |
| Non-users | 360 | ref. |
| Users | 952 | 1.15 (1.01–1.32) |
| ACE inhibitors | 415 | 1.19 (1.02–1.39) |
| ATr blockers | 208 | 1.1(0.89–1.36) |
| Beta-blockers | 559 | 1.04 (0.91–1.2) |
| Calcium channel blockers | 369 | 0.97 (0.83–1.14) |
| Diuretics | 336 | 0.91 (0.75–1.1) |
| **Metastatic disease** | | |
| Non-users | 166 | ref. |
| Users | 381 | 1.08 (0.88–1.31) |
| ACE inhibitors | 189 | 1.09 (0.87–1.38) |
| ATr blockers | 87 | 0.94 (0.66–1.33) |
| Beta-blockers | 246 | 1 (0.82–1.22) |
| Calcium channel blockers | 158 | 1.03 (0.81–1.3) |
| Diuretics | 136 | 0.96 (0.73–1.27) |

failures as compared to non-users. However, the study consisted only of 62 patients and the mean follow-up time was 29 months. Alashkham et al. [26] showed that biochemical recurrence was decreased after curative-intent radiotherapy with hormone treatment among users of RAS-inhibiting drugs as compared to non-users. Our study supports these findings, although we did not see any reduction in the risk of initiation of ADT therapy among users of RAS-inhibiting drugs.

Results from analysis of pre-diagnostic use of antihypertensive drugs and PCa-specific survival was in line with the results of post-diagnostic use: overall antihypertensive drug use was associated with an increased risk for PCa-specific death, with decreased mortality among users of ATr blockers and an increase in users of diuretics. Furthermore, pre-diagnostic use of antihypertensive drugs was associated with increased all-cause mortality. This was also seen with several different drug groups: ACE inhibitors, beta-blockers, and calcium channel blockers, but not simply ATr blockers. These results underline that, in general, users of antihypertensive drugs are at an increased risk of death as compared to non-users, but the association is different for users of ATr blockers. A similar trend was evident when the association of antihypertensive drugs and CVD-specific deaths was analyzed. The results were similar when deaths due to CVD were taken into account using competing risk analysis, suggesting that the risk association with PCa survival was not affected by the higher risk for CVD death among antihypertensive drug users compared to non-users.

The reason why specifically it is the use of ATr blockers that may improve PCa-specific survival is not clear. In our study, users of ATr blockers had a smaller proportion of metastatic cancer cases, a lower likelihood of initiation of ADT therapy, and lower PCa mortality as compared to patients in the other groups. This could be due to selective prescribing of this drug group to healthier men than other antihypertensive drugs, or due to genuine effects of ATr blockers on PCa. Our results are in line with a recently published study, where the use of ATr blockers increased PCa-specific survival after radical prostatectomy [18]. ATr blockers are antagonists of angiotensin II receptor type 1, which mediates the classical angiotensin II-related effects; vasoconstriction, fluid volume homeostasis, cell proliferation, and fibrosis [27]. Other RAS-inhibiting drugs, ACE inhibitors, inhibit the main angiotensin II forming enzyme, thus both drug groups block angiotensin II-mediated functions, albeit at different sites in the pathway. ATr blockers may have some yet unknown mechanism of action impacting on PCa development and progression. Nevertheless, the possible role of RAS in development and progression of cancer is under investigation [7,28,29].

Diuretics are commonly used in management of edema, which is common in advanced cancer. Thus, loop diuretics (furosemide) and spironolactone, which are used in management of edema rather than hypertension, were excluded from this analysis. Still, the use of diuretics was associated with an increased risk for PCa death despite lack of association with a higher stage cancer, such metastatic or risk group 2 cases, which might be expected if increased the association could be explained by the treatment of end-stage cancer-related problems. In agreement, Holmes et al. [11] reported an increased risk for PCa-specific deaths among thiazide diuretics users. In our study, we did not evaluate the role of separate diuretic classes.

We did not find any association of pre- and post-diagnostic use of beta-blockers with prostate cancer-specific mortality. Previous studies have reported controversial results [14–17]. However, in analyses based on time-dependent variables, as in our study, no association has been found [30, 31]. However, the use of beta-blockers was associated with a slightly increased risk for ADT initiation, pointing to some association with tumor progression. In contrast to our findings, in a Norwegian population-based cohort, Grytli et al. [16] showed that the use of beta-blockers during ADT-therapy seemed to decrease the risk of PCa-specific death.

However, that study was analyzed without time-dependent variables. Nevertheless, the link between beta-blockers and PCa outcomes remains unclear.

A recently published study [32] investigated the role of antihypertensive drug use on PCa-specific mortality in gonadotropin-releasing hormone agonist users. They showed a slightly increased PCa mortality in users of blood pressure lowering drugs. We did not have information on blood pressure levels and were thus unable to evaluate the role of hypertension in this risk association.

Our study has several strengths. The Finnish prescription database is comprehensive and accurate; for example, all reimbursements for purchases of antihypertensive drugs are recorded, and thus we had nearly complete coverage of our subjects' drug purchases. Our follow-up time was long enough to enable modeling cumulative simultaneous use of multiple drugs. We also had comprehensive information of clinical features of prostate cancer, such as Gleason grade and TNM stage. In addition, information on PCa deaths was collected from an accurate and reliable nationwide, population-based database that has been validated by an independent cause of death committee; the validity of information of PCa-specific deaths between cause of death registry and recording in patients' medical files was compared and 97.7% agreement (kappa = 0.95) between them was found [33]. Furthermore, statistical power in our cohort is sufficient to detect clinically meaningful survival differences as our sample size was sufficient to detect 3.5% survival difference between medication users and non-users with 80% power and risk for type I error being 0.05.

Our study also has some limitations. We did not have information on blood pressure levels of the subjects or indication for antihypertensive drugs. In addition, data of some background information, which might be shared risk factors for prostate cancer death and hypertension, such as smoking habits, were missing. However, we were able to adjust the analysis for the use of other drugs which might have impacted on the development and progression of PCa, such as statins [34]. Our drug use data is based on reimbursement for drug purchases. Thus, we do not know whether the subject actually used the purchased drugs. It should be noted that most likely the users of antihypertensive drugs have more comorbidities and are at increased risk of death in general as compared to non-users, possibly affecting also PCa-specific mortality. Furthermore, different antihypertensive drugs are described based on comorbidities and the health status of the patients which might bias the results.

In conclusion, the use of antihypertensive drugs was associated with an increased risk for prostate cancer-specific and all-cause mortality. When the analysis was done separately for different drug groups, both pre- and post- diagnostic use of RAS inhibiting drugs, especially ATr blockers, associated with improved PCa survival in a dose-dependent manner, whereas the use of diuretics associated with poorer survival. The results might be affected by systematic differences between the users and non-users. Nevertheless, our findings support previous reports about the beneficial effects of RAS-inhibition on PCa progression. The role and potential benefits of RAS inhibition in PCa should be examined further.

## Supporting information

**S1 Table. Risk for pre-diagnostic use of antihypertensive use on prostate cancer-specific mortality.** Users stratified by intensity of use into tertiles (low, medium and high dose). ACE inhibitors = angiotensin-converting enzyme inhibitors and Atr blockers = angiotensin II type 1 receptor blockers.
(XLSX)

**S2 Table. Risk of prostate cancer-specific mortality according to the use of antihypertensive drugs after diagnosis of PCa.** Users stratified to tertiles by cumulative years of the use

(A) and cumulative DDD-values (B). ACE inhibitors = angiotensin-converting enzyme inhibitors, ATr blockers = angiotensin II type 1 receptor blockers, HR (95% CI) = hazard ratio with 95% confidence interval.
(XLSX)

**S3 Table. Lagtime analysis using 1-year and 3-year lagtimes of use of different antihypertensive drugs on PCa-specific mortality.** Cox regression model was adjusted with age, FinRSPC trial arm (screening arm and control arm), year of diagnosis, cancer clinical characteristics (T stage, metastasis, and Gleason grade), Charlson comorbidity index, use of statins, antidiabetic drugs, anticoagulants, 5-alpha-reductase inhibitors, aspirin and other NSAIDs.
(XLSX)

**S4 Table. Risk of initiate hormonal treatment for prostate cancer by the use of antihypertensive drugs, users stratified by cumulative intensity of the use (DDD values/years of the use).** ACE inhibitors = angiotensin-converting enzyme inhibitor, ATr = angiotensin II type 1 receptor blockers.
(XLSX)

**S5 Table. Sensitivity analysis of number of used drugs and combination of different drug groups on PCa-specific death.**
(XLSX)

## Author Contributions

**Conceptualization:** Teemu J. Murtola, Kimmo Taari, Teuvo L. J. Tammela, Anssi Auvinen.

**Data curation:** Aino Siltari, Teemu J. Murtola.

**Formal analysis:** Aino Siltari.

**Funding acquisition:** Teemu J. Murtola, Anssi Auvinen.

**Investigation:** Aino Siltari, Teemu J. Murtola, Kimmo Taari, Teuvo L. J. Tammela, Anssi Auvinen.

**Methodology:** Aino Siltari, Teemu J. Murtola.

**Project administration:** Kirsi Talala, Anssi Auvinen.

**Resources:** Kirsi Talala.

**Supervision:** Teemu J. Murtola, Anssi Auvinen.

**Validation:** Teemu J. Murtola, Kirsi Talala, Anssi Auvinen.

**Visualization:** Aino Siltari.

**Writing – original draft:** Aino Siltari.

**Writing – review & editing:** Aino Siltari, Teemu J. Murtola, Kirsi Talala, Kimmo Taari, Teuvo L. J. Tammela, Anssi Auvinen.

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
