## [Decision Letter · Decision Letter 0]

21 Apr 2020

PONE-D-20-06329

Antihypertensive drug use and prostate cancer-specific mortality in Finnish men

PLOS ONE

Dear Dr Siltari,

Thank you for submitting your manuscript to PLOS ONE. After careful consideration, we feel that it has merit but does not fully meet PLOS ONE’s publication criteria as it currently stands. Therefore, we invite you to submit a revised version of the manuscript that addresses the points raised during the review process.

We would appreciate receiving your revised manuscript by Jun 05 2020 11:59PM. To enhance the reproducibility of your results, we recommend that if applicable you deposit your laboratory protocols in protocols.io, where a protocol can be assigned its own identifier (DOI) such that it can be cited independently in the future. For instructions see: http://journals.plos.org/plosone/s/submission-guidelines#loc-laboratory-protocols

We look forward to receiving your revised manuscript.

Kind regards,

Jian Gu

Academic Editor

PLOS ONE

2. In your ethics statement in the manuscript and in the online submission form, please provide additional information about the patient records used in your retrospective study. Specifically, please ensure that you have discussed whether all data were fully anonymized before you accessed them.

Reviewers' comments:

Reviewer's Responses to Questions

**Comments to the Author**

1. Is the manuscript technically sound, and do the data support the conclusions?

Reviewer #1: Yes

Reviewer #2: Yes

2. Has the statistical analysis been performed appropriately and rigorously? 

Reviewer #1: Yes

Reviewer #2: Yes

3. Have the authors made all data underlying the findings in their manuscript fully available?

Reviewer #1: Yes

Reviewer #2: Yes

4. Is the manuscript presented in an intelligible fashion and written in standard English?

Reviewer #1: Yes

Reviewer #2: Yes

5. Review Comments to the Author

Reviewer #1: This is an interesting study but I have a few minor comments/suggestions:

1) I do not understand why spironolactone is excluded. Three observational studies (UK, HongKong, Sweden) have now shown that this is potentially inversely associated with prostate cancer risk. Hence, it would be of interest to also see what happens with pca progression and death.

2) The competing risk analyses are very minimalistic. There is no description of CVD deaths, history based on use of anti-hypertensive. What about showing the cumulative incidence graphs for pca specific death, overall death, cvd death - so see a composition of the deaths based on exposure value?

Potentially the results might be over-interpreted if this information is not completely shown.

Reviewer #2: The article “Antihypertensive drug use and prostate cancer-specific mortality in Finnish men” by Siltari et al, who aimed to evaluate both pre- and post-diagnosis use of antihypertensive regimens on prostate cancer-specific survival and the initiation of androgen deprivation therapy, is of clinical and public health importance. Via a robust analysis, the authors found anti-cancer effect of angiotensin II type1 receptor blockers on prostate cancer prognosis. I here raise several comments for further minor revision.

1- The power to detect significance should be addressed.

2- The dosage of antihypertensive regimens should be considered.

3- It is of added interest to see whether this is a dose-response association between the number of antihypertensive drugs and the prognosis risk of prostate cancer.

6. PLOS authors have the option to publish the peer review history of their article (what does this mean?). If published, this will include your full peer review and any attached files.

Reviewer #1: No

Reviewer #2: Yes: Wenquan Niu

---

## [Author Response · Author response to Decision Letter 0]

18 May 2020

Response: We have now gone through our manuscript and ensured that it fits PLOS ONE style requirements.

2. In your ethics statement in the manuscript and in the online submission form, please provide additional information about the patient records used in your retrospective study. Specifically, please ensure that you have discussed whether all data were fully anonymized before you accessed them.

Response: We have now added information about the patient records to the manuscript and to our ethics statement. We used unique personal identification numbers as the key to find and link information across registries. After linkage, however, identification numbers were replaced with study numbers before analysis. 

Response: We have now included results of this analysis in the supplemental material (S4 table).

Response: Sharing individual-level data, even in pseudonymized form, is not possible according to the current Finnish regulation regarding privacy, data protection and EU-level GDPR. Full anonymization of the data is not feasible, because at individual level, there would be numerous unique records if all variables used in the analyses were to be retained (even if collapsing discrete variables and categorizing continuous variables). Permission for data access was applied from Finnish institute of health and welfare and data access can be applied from there (kirjaamo@thl.fi)

Even though we cannot share our individual-level data, if required we can create a metadata document which includes information about all variables with summary values which we have in our data sets and add it to the supplementary materials. 

Response to reviewers’ comments:

Reviewer #1: This is an interesting study but I have a few minor comments/suggestions:

Comment: 1) I do not understand why spironolactone is excluded. Three observational studies (UK, HongKong, Sweden) have now shown that this is potentially inversely associated with prostate cancer risk. Hence, it would be of interest to also see what happens with pca progression and death.

Response: We excluded spironolactone as it is commonly used to treat edema which is common in advanced prostate cancer cases. In Finland spironolactone is used to treat edemas rather than used as an antihypertensive drug, which causes powerful confounding when studying cancer mortality as edema is quite in advanced cancer.

Concordantly, if we combine spironolactone in our analysis within the diuretics group, use of diuretics is strongly associated with increased prostate cancer-specific mortality, thus confirming confounding by indication.

Comment: 2) The competing risk analyses are very minimalistic. There is no description of CVD deaths, history based on use of anti-hypertensive. What about showing the cumulative incidence graphs for pca specific death, overall death, cvd death - so see a composition of the deaths based on exposure value?

Potentially the results might be over-interpreted if this information is not completely shown.

Response:

We understand reviewer’s concern about possible bias by competing risks of deaths and thus over-interpretation. We have now plotted cumulative hazard plots (see attached response to reviewers document) which indicate that cumulative hazards are parallel for PCa-specific, CVD and all cause death. Thus, the observed association between antihypertensive drugs and prostate cancer survival is not likely to be mitigated by differing association with CVD mortality. Analysis was done for prediagnostic antihypertensive use and was adjusted similar as other analysis in our study.

Reviewer #2: The article “Antihypertensive drug use and prostate cancer-specific mortality in Finnish men” by Siltari et al, who aimed to evaluate both pre- and post-diagnosis use of antihypertensive regimens on prostate cancer-specific survival and the initiation of androgen deprivation therapy, is of clinical and public health importance. Via a robust analysis, the authors found anti-cancer effect of angiotensin II type1 receptor blockers on prostate cancer prognosis. I here raise several comments for further minor revision.

Comment: 1- The power to detect significance should be addressed.

Response: We agree that power to detect significant difference is important. Thus, we conducted power calculation based on overall deaths of antihypertensive drug users vs. non-users (622 (33.2%) vs. 2143 (33.6%)) in our cohort. Assuming 80% statistical power and risk for type I error as 0.05, our sample size is sufficiently large to detect 3.5% difference between medication users and non-users. Thus, we conclude that power in our cohort is sufficient as the observed risk differences were greater than this. We have added this point to the Discussion, see lines 369-372:

“Furthermore, statistical power in our cohort is sufficient to detect clinically meaningful survival differences as our sample size was sufficient to detect 3.5% survival difference between medication users and non-users with 80% power and risk for type I error being 0.05.”

Comment: 2- The dosage of antihypertensive regimens should be considered.

Response: We have analyzed dose-response using annual cumulative DDD-values and intensity of use (cumulative DDD-values/years of total use) separately for each drug group. Annual DDD-values and intensities are stratified by tertiles to estimate survival associations in low, medium and high exposure. These results are reported in Table 4 and tables S1 and S2.

Comment: 3- It is of added interest to see whether this is a dose-response association between the number of antihypertensive drugs and the prognosis risk of prostate cancer.

Response: We have performed suggested additional analysis where number of used drugs and different combinations of drugs (e.g. AT receptor blockers and diuretics) were included as variables. We did not find any clear indication that number of drugs or different drug combination would impact on our results, with possible exception of diuretics + betablockers, users of which had higher risk of prostate cancer death compared to non-users than other drug combinations. Table is now added to supplemental material (S5 table).

These findings have been added to the manuscript:

Materials and Methods, lines 132-133:

“We also conducted sensitivity analysis by number of used drug groups (one, two, or three or more) and by combinations of different drug groups (e.g. ATr blockers + diuretics).”

Results, lines 229-235:

“Sensitivity analysis by number of used drugs did not modify the results as risk for PCa-specific death was increased similarly despite number of drug groups in use (S5 table). Some drug combinations showed slightly different survival associations compared to main analysis; combination of beta-blockers and diuretics were associated with lowered risk of PCa death compared non-users (HR 0.74, 95% CI 0.57-0.96) whereas diuretics and beta-blockers in combination were associated with increased risk (HR 2.36, 95% CI 1.89-2.94). Other drug group combination did not show significant associations with PCa-specific survival (S5 table).”

---

## [Editor Report · Decision Letter 1]

22 May 2020

Antihypertensive drug use and prostate cancer-specific mortality in Finnish men

PONE-D-20-06329R1

Dear Dr. Siltari,

We are pleased to inform you that your manuscript has been judged scientifically suitable for publication and will be formally accepted for publication once it complies with all outstanding technical requirements.

With kind regards,

Jian Gu

Academic Editor

PLOS ONE
---

## [Editor Report · Acceptance letter]

15 Jun 2020

PONE-D-20-06329R1 

Antihypertensive drug use and prostate cancer-specific mortality in Finnish men 

Dear Dr. Siltari:

I'm pleased to inform you that your manuscript has been deemed suitable for publication in PLOS ONE. Congratulations! Your manuscript is now with our production department. 

Kind regards, 

on behalf of

Dr. Jian Gu 

Academic Editor

PLOS ONE